# Printed and Flexible ECG Electrodes Attached to the Steering Wheel for Continuous Health Monitoring during Driving

**DOI:** 10.3390/s22114198

**Published:** 2022-05-31

**Authors:** Joana M. Warnecke, Nagarajan Ganapathy, Eugen Koch, Andreas Dietzel, Maximilian Flormann, Roman Henze, Thomas M. Deserno

**Affiliations:** 1Peter L. Reichertz Institute for Medical Informatics of TU Braunschweig and Hannover Medical School, 38106 Braunschweig, Germany; nagarajan.ganapathy@plri.de (N.G.); thomas.deserno@plri.de (T.M.D.); 2Institute of Microtechnology (IMT), TU Braunschweig, 38124 Braunschweig, Germany; eugen.koch@tu-braunschweig.de (E.K.); a.dietzel@tu-braunschweig.de (A.D.); 3Institute of Automotive Engineering, TU Braunschweig, 38106 Braunschweig, Germany; m.flormann@tu-braunschweig.de (M.F.); r.henze@tu-braunschweig.de (R.H.)

**Keywords:** printed electrodes, digital prevention, smart car, health monitoring

## Abstract

Continuous health monitoring in a vehicle enables the earlier detection of symptoms of cardiovascular diseases. In this work, we designed flexible and thin electrodes made of polyurethane for long-term electrocardiogram (ECG) monitoring while driving. We determined the time for reliable ECG recording to evaluate the effectiveness of the electrodes. We recorded data from 19 subjects under four scenarios: rest, city, highway, and rural. The recording time was five min for rest and 15 min for the other scenarios. The total recording (950 min) is publicly available under a CC BY-ND 4.0 license. We used the simultaneous truth and performance level estimation (STAPLE) algorithm to detect the position of R-waves. Then, we derived the RR intervals to compare the estimated heart rate with the ground truth, which we obtained from ECG electrodes on the chest. We calculated the signal-to-noise ratio (SNR) and averaged it for the different scenarios. Highway had the lowest SNR (−6.69 dB) and rural had the highest (−6.80 dB). The usable time of the steering wheel was 42.46% (city), 46.67% (highway), and 47.72% (rural). This indicates that steering-wheel-based ECG recording is feasible and delivers reliable recordings from about 45.62% of the driving time. In summary, the developed electrodes allow continuous in-vehicle heart rate monitoring, and our publicly available recordings provide the opportunity to apply more sophisticated data analytics.

## 1. Introduction

Health monitoring in private spaces such as vehicles enables digital prevention [1]. According to the World Health Organization (WHO), cardiovascular diseases cause 17 million deaths per year worldwide [2]. Symptoms such as atrial fibrillation (AF) or other types of arrhythmia can be spread over a longer period [3]. AF increases the risk of stroke, and the onset of AF is often undiagnosed [4]. Daily and continuous monitoring enable early detection of symptoms and early intervention, improving therapeutic outcomes and decreasing mortality rates [5]. Moreover, it supports physicians in diagnosing the spread of these symptoms in a timely manner [4].

In-vehicle monitoring has several advantages. On average, a person spends 30 min per day in a vehicle [6]. Unobtrusive measurements do not require any additional action of the driver. Furthermore, the layout in a vehicle is rather static, as compared to that in an apartment. Therefore, a medical check-up can be integrated into daily life. An electrocardiogram (ECG) shows several diagnostic parameters, e.g., the heart rate (HR), as well as pathological changes and cardiac arrhythmia [7]. Some publications have reported on the integration of ECG electrodes into a steering wheel. In 2007, Lee et al. used dry electrodes made of copper tape [8]. Baek et al. used copper plates with a length of a few centimeters [9].

Vavrinský et al. mounted aluminum macroelectrodes to the steering wheel [10]. Shin et al. used conductive fabric electrodes for ECG measurement, and also recorded a photoplethysmogram (PPG) [11]. Heuer et al. developed textile capacitive electrodes placed in the seat [12]. Tomimori et al. integrated metal electrodes into the steering wheel [13]. Gomez-Clapers and Casanella used a wireless steering wheel with four dry stainless steel ECG electrodes [14], but did not integrate their system into a car. Silva et al. used dry Ag/AgCl electrodes and electrolytes [15]. Jung et al. applied conductive fabric electrodes on the steering wheel [16]. In 2019, Cassani et al. placed eight electrodes on the steering wheel of a driving simulator [17]. Babusiak et al. developed a new steering wheel design with two integrated ECG electrodes, which required the replacement of the steering wheel [18].

Most of these articles reported a low signal-to-noise ratio (SNR) for steering-wheel-based ECG recordings. However, features such as T-, R-, and P-waves were visible. The design of electrodes for the steering wheel has specific challenges, especially since the placement of electrodes should not obstruct the driver during driving. Nonetheless, existing electrodes are inflexible and, therefore, have a small surface, which limits the hand positions of the driver for recording the ECG. Furthermore, flat electrodes are advantageous. Research still lacks an optimal design for electrodes.

Moreover, ECG signal processing methods already exist to recognize cardiovascular diseases (CVDs). In 2016, Ahmed et al. published a survey for ECG signal preprocessing [19]. They discussed CVD recognition based on preprocessing, QRS detection, feature extraction, and classification, and pointed out that baseline wander and noise removal can be performed adaptively [20], or using median [21] and bandpass filters [22]. QRS detection often is based on derivatives [23], wavelet transform [24], neural networks [25], or hidden Markov models [26]. Ahmed et al. categorized the features for classification in time-, frequency-, and time–frequency-based features [19]. In the literature, the variety of classifiers ranges from neural networks [27] to k-nearest neighbors [28]. In 2021, Hua et al. developed an approach for information divergences based on divergence-based matrix information geometry detectors [29]. Moreover, Jain et al. implemented an application-specific integrated circuit (ASIC) on a smartphone to detect symptoms in an energy-efficient manner [30]. Furthermore, several works have published multisensor or multichannel approaches [31,32].

However, we focused on in-vehicle monitoring and a simple approach for the classification of reliable and unreliable signals. To investigate whether continuous health monitoring with steering-wheel-based ECG recordings is feasible, we focused on designing electrodes and answering the question “What percentage of the driving time results in reliable ECG signals?”.

In Section 2, we explain the development process of the ECG electrodes, the sensor system for the recording, the experimental design, how a person can access the recorded data, and the generation of the ground truth. Section 3 presents a comparison of the SNRs between different driving scenarios and the steering wheel ground-truth data. We discuss the applied methods and the results in Section 4. Our paper ends with a conclusion (Section 5).

## 2. Materials and Methods

### 2.1. Electrode Design

To derive the positions and shape of the electrodes, we generated a 3D model of the steering wheel of our research vehicle (VW Tiguan, Volkswagen, Wolfsburg, Germany). We converted this 3D model into a 2D shape to derive the 2D electrode shape (Figure 1).

We used a thermoplastic polyurethane (TPU) substrate with screen-printed silver paste on top. The base film (IntexarTM TE-11C, DuPont, Wilmington, NC, USA) had three different layers: high-recovery TPU, melt-adhesive TPU, and a temperature-stable carrier [33]. The silver paste (Intexar^TM^ PE-874, DuPont, Wilmington, NC, USA) has a high conductivity [34]. Furthermore, these materials are stretchable and flexible. Both materials were designed for wearable applications and, therefore, exhibit inherent stretchability. The fabricated screen-printing frame had a size of A3+ with 39 threads per cm (screen-printing frame 39T of size 61 cm × 51 cm, Siebdruckversand, Magdeburg, Germany). The mesh size of each thread was 172 µm, and the fabric thickness was 142 µm. A sieve filtered the silver paste. The electrodes were dried at 130 °C for 15 min in the oven (LCD-1, Despatch Industries, Minneapolis, MN, USA). The electrode shape has two larger circles that are connected by a curved trace. To increase comfort during driving and to maintain maximum contact, the electrodes cover the lower half of the steering wheel (Figure 2).

### 2.2. Sensor System

On the bottom of the electrodes, we placed an adhesive electrode (AgCl electrodes, Covidien, Dublin, Ireland) to establish a connection to the ECG sensor (Explorer Kit, Biosignalplux, Lisbon, Portugal) and the electrodes (Figure 2).

Another ECG sensor (Explorer Kit, Biosignalplux, Lisbon, Portugal) was used to record the reference ECG with adhesive electrodes, which we attached to three positions of the thorax: the right arm (positive), left arm (ground), and left leg (negative). The sampling rate of both ECG sensors was 500 Hz. Before each recording, we conducted a pretest to ensure that the electrodes were in the correct position.

We synchronized the timing of our recordings using a channel hub (Explorer Kit, Biosignalplux, Lisbon, Portugal). We mounted the channel hub at the backside of the moving steering wheel, and wirelessly connected it to a single-board computer (Raspberry Pi, model 4, 8 GB RAM, Raspberry Pi Foundation, Cambridge, UK) via Bluetooth. For data access, we used the Python application programming interface [35].

### 2.3. Experimental Design

We recorded ECG data from 19 healthy subjects with four scenarios (Figure 2):

*Rest*: the engine is on, and the subject sits in a comfortable position while moving the steering wheel.

*City*: the subject drives in a city, with a maximum speed of 50 km/h.

*Highway*: the subject drives on a highway, with a speed limit of 130 km/h.

*Rural*: the subject drives in a rural area. The route passes through villages and a railway crossing. The speed limit for the rural roads is 100 km/h.

The recording time was 5 min for rest and 15 min for city, highway, and rural. We prescribed the driving routes to ensure comparability between the recordings. The data were recorded in accordance with the Helsinki Declaration. All subjects signed a consent form.

### 2.4. Database

As the ECG signals do not enable us to derive the identity of the subject, we published the data anonymously via the library of TU Braunschweig with the license CC BY-ND 4.0 (s. Data Availability Statement and link: https://doi.org/10.24355/dbbs.084-202203170707-0, accessed on 28 May 2022). The data repository includes the following (see Appendix A):ECG reference signal;ECG signal acquired from the electrodes on the steering wheel;Metadata (e.g., age, height, weight, gender).

### 2.5. Ground Truth

We applied the simultaneous truth and performance level estimation (STAPLE) algorithm to determine the position of the R-waves and to derive the HR of the reference ECG [36]. Our STAPLE was composed of nine state-of-the-art algorithms, developed by Pan and Tompkins [37], Chernenko [38], Arzeno et al. [23], Manikandan et al. [39], Lentini et al. [40], Sartor et al. [41], Liu et al. [42], Arteaga-Falconi et al. [43], and Khamis et al. [44]. Based on a weighted majority voting, STAPLE was used to determine the algorithms’ performances and the positions of the R-waves. We implemented and execute STAPLE using MATLAB (version R2021a) [36]. The SNR was used to calculate the power of the signal divided by the sum of the power of noise:(1)$$SNR=10\;lg\left(\frac{P_{Signal}}{P_{Noise}}\right)dB$$

We calculated the SNR of the steering wheel ECG based on the reference ECG frequency, and used the MATLAB function periodogram to estimate the power spectral density [45]. For each R-wave position in the reference ECG, we computed the duration of the RR interval (RRI). We calculated the HR as HR=60/RRI [46].

### 2.6. Rules for Classification of Usable and Unusable Signals

For the preprocessing, we used a Butterworth bandpass (Figure 3) to remove baseline wander and noise, with cutoff frequencies of 0.5 Hz and 25 Hz [47]. We considered the RRI as a time-domain feature. We classified signal segments as usable based on two thresholds: The first considers the median of the reference HR, which forms a tube around the signal to exclude excessively high amplitudes. The normal range for HR is between 60 bpm and 100 bpm [48]. Based on the work of Langendorf et al. [35], we labeled a segment as unreliable if the RRI changed by more than 0.08 s. The reference ECG, however, was considered as reliable disregarding this threshold.

## 3. Results

The steering wheel ECG inherits characteristic features—e.g., P- and T-waves, as well as the QRS complex—but the amplitudes of the R-waves are smaller as compared to the reference ECG (Figure 3).

The average SNR for the scenario rest was the lowest, at −5.49 dB. In the other three scenarios, the average SNR was −6.75 dB, −6.69 dB, and −6.80 dB for city, highway, and rural, respectively. As in the work of Allen et al. [49], a raincloud plot (Figure 4) was used to visualize the distribution of the SNR (left-hand side) and individual data points (right-hand side). The negative values of the SNRs indicate that the steering wheel ECG is noisier than the reference signal.

For the rest scenario, we recorded 5 min for each of 19 test subjects = 95 min. The recording followed the reference, and only a few red dots on the *x*-axis indicate unreliable steering wheel measures (Figure 5). In total, 53 min (55.79%) were usable for further analysis.

As shown in the following figures, we analyzed the recordings of subject 2209, arbitrarily chosen as an average example. The SNRs from this test person were 0.84 dB (rest), −7.25 dB (city), −7.78 dB (highway), and −6.99 dB (rural), and were neither the worst nor the best values.

The SNR in the city was lower. Starting and stopping driving due to traffic lights or turns causes motion artifacts when the hands are detached from the steering wheel during steering movements (Figure 6). For the city scenario, we recorded 15 min for each subject, yielding 285 min in total. From these recordings, 121 min (42.46%) were reliable.

The highway scenario has additional vibrations due to the higher speed, and the drivers detach their hands from the steering wheel when indicting a lane change. We observed that the HR was lower as compared to driving in the city (Figure 7). From the 285 min of recording, 133 min (46.67%) were useable for further analysis.

On rural roads, the SNR was lowest (Figure 8). In total, 136 min out of 285 min (47.72%) were labeled as usable.

## 4. Discussion

Continuous health monitoring is relevant for detecting cardiovascular diseases in their early stages [5]. The proposed ECG electrodes in the existing literature are static electrodes with a smaller surface [8,10,18]. We developed flexible and thin electrodes with a taller surface that can be printed easily and cheaply. These electrodes do not disturb the driver’s behavior and enable subsequent in-vehicle integration.

With our electrodes, we captured a reliable HR from the ECG recordings for about 25% of the driving time. This proportion was dependent on several factors, e.g., skin properties, scenario, and driving route. We were surprised by the rather low reliability for the rest scenario (55.79%). This might have been due to the movements of the test subjects. Moreover, the SNR for the rest scenario had a high variance between the different test subjects.

As a limitation, our research car was equipped with an automatic transmission. This prevents dropouts from gear-level shifts. However, our recordings show a proof of concept for electrodes made of polyurethane. Recordings with further subjects and for a longer duration may confirm our findings. Furthermore, the driving scenario impacts reliability only slightly. There was not much difference between the city, highway, and rural scenarios. This might have been due to the comparable SNRs.

Another limitation of our work is that we focused on the development of thin and flexible electrodes, in-vehicle data recording, and classification of reliable and unreliable signal periods, but not on advanced signal analysis. To foster further data analytics with machine learning or other rule-based algorithms, our data are available at https://doi.org/10.24355/dbbs.084-202203170707-0 (accessed on 28 May 2022). The usage of machine learning approaches—e.g., support-vector machines, k-nearest neighbors, and convolutional neural networks (CNNs)—could lead to improved data analysis. This integration could increase the usable recording time and allows data tracking over extended periods of time to see if heart performance has changed from a baseline.

We consider 45.62% as a lower bound. Integrating additional sensors such as photoplethysmograms (PPGs) [11], cameras [50], radar [51], capacitive ECG [52], and near-infrared cameras [53] will improve the length of reliable measures, since the redundant system can instantaneously choose the sensor yielding the best data quality [54]. Furthermore, the shape and size of the driver’s hands have an impact. Skin properties such as temperature and humidity—resulting from physical and mental activities, as well as from sweating—differ between individuals [55,56]. In our calculations for the SNR, we could not identify a change in the SNR between the beginning and end of a longer driving time. This could occur if sweat accumulates on the electrodes from the hands after a long drive. However, we expect a better signal from sweating hands, as sweat is composed of electrolytes and increases the conductivity [55].

Furthermore, the subjects have different driving styles and handgrips [57]. Future work should also address the integration of biomedical sensors in the CAN-BUS system and its transmission via 5G [58].

## 5. Conclusions

Continuous health monitoring is relevant to detecting cardiovascular diseases in their early stages [5]. Our results show that ECG electrodes attached to the steering wheel can capture the ECG reliably for about 45.62% of the driving time. If the average driving time is 30 min per day, we could thus obtain 13.67 min of heart rate recording per day.

This would impact the early detection and prevention of stroke and other signals for specific periods. Better materials and different shapes of the electrodes might increase the reliability.

## Figures and Tables

**Figure 1 sensors-22-04198-f001:**
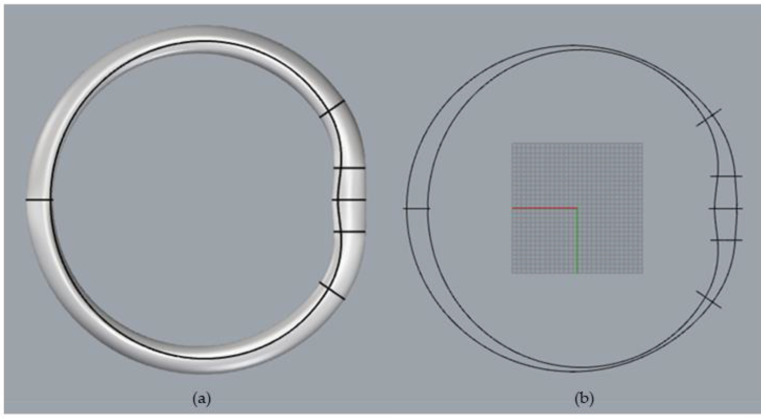
Design process: (**a**) 3D model of the steering wheel, (**b**) 2D model.

**Figure 2 sensors-22-04198-f002:**
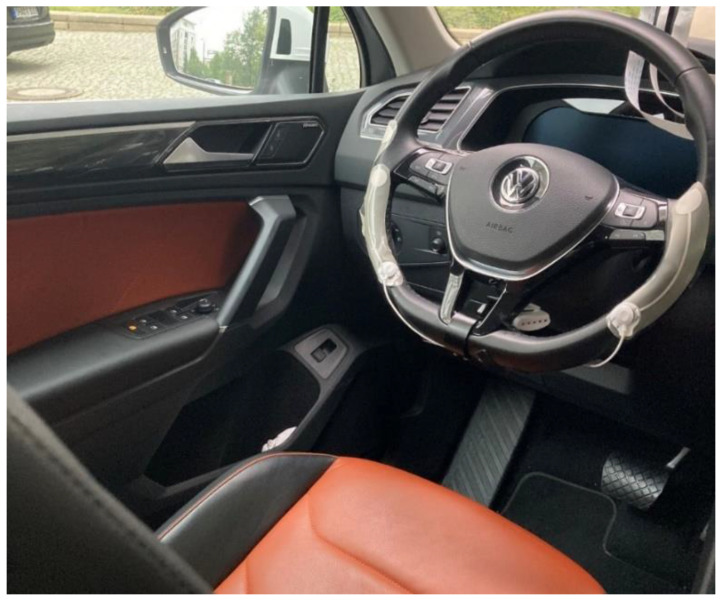
Recording system with the ECG electrodes in the research car.

**Figure 3 sensors-22-04198-f003:**
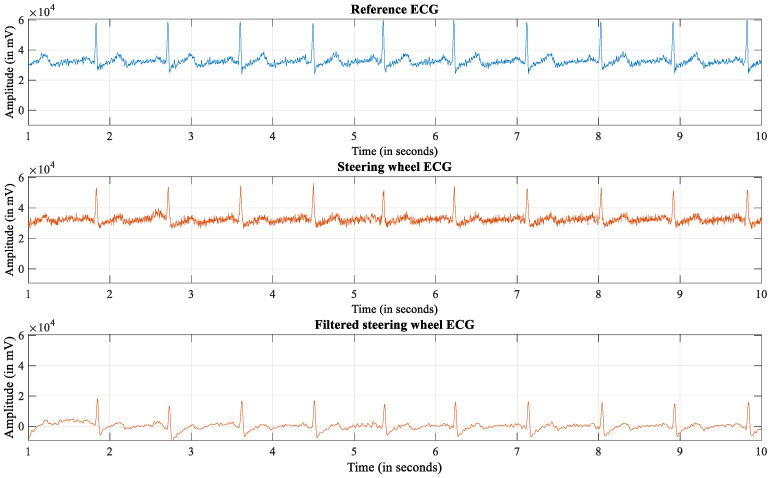
ECG recordings during rest.

**Figure 4 sensors-22-04198-f004:**
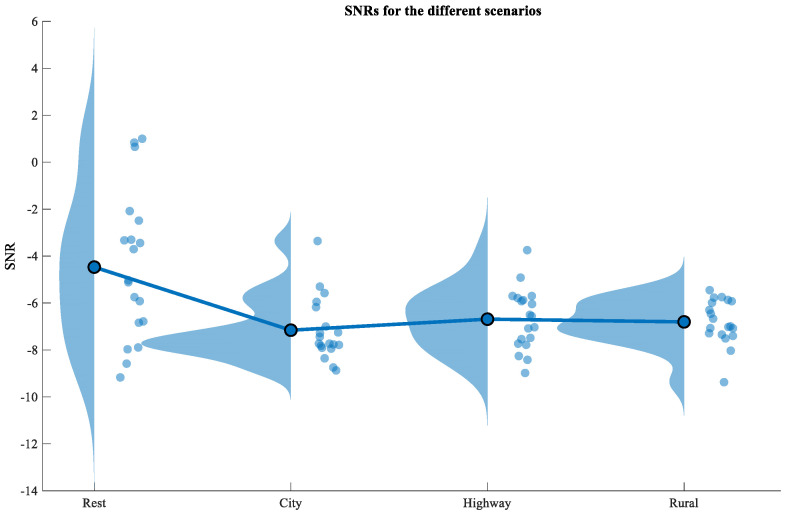
Distribution of the SNR for the different scenarios.

**Figure 5 sensors-22-04198-f005:**
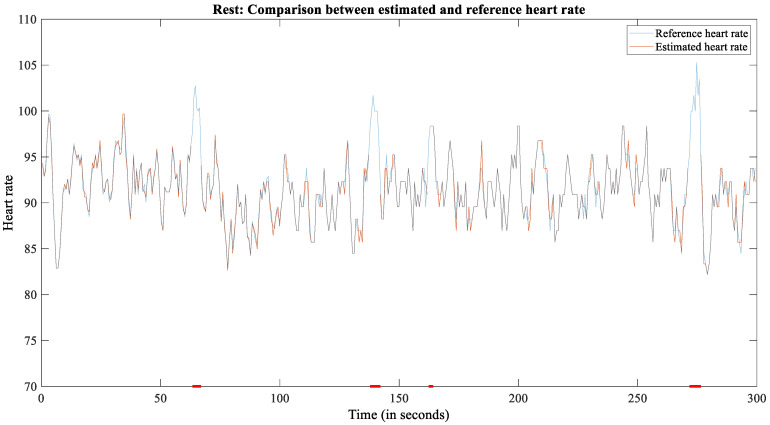
The plot of reference and estimated HR for subject 2209 during rest.

**Figure 6 sensors-22-04198-f006:**
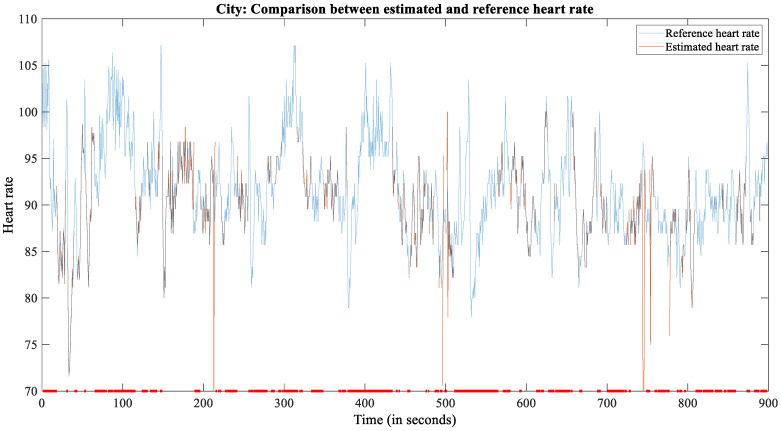
The plot of reference and estimated HR for subject 2209 during the city scenario.

**Figure 7 sensors-22-04198-f007:**
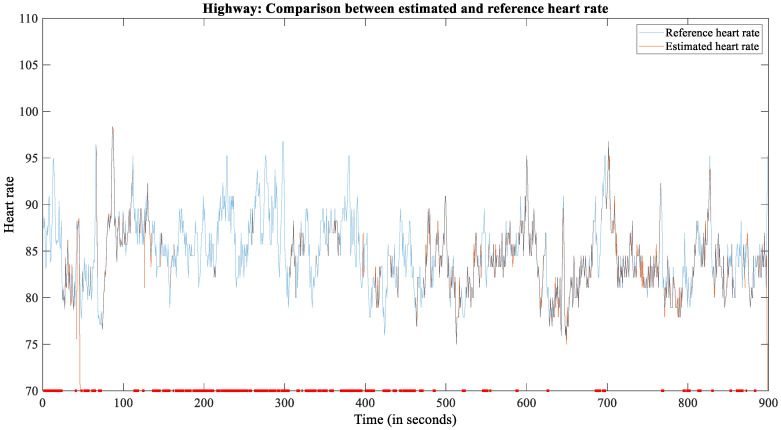
The plot of reference and estimated HR for subject 2209 during the highway scenario.

**Figure 8 sensors-22-04198-f008:**
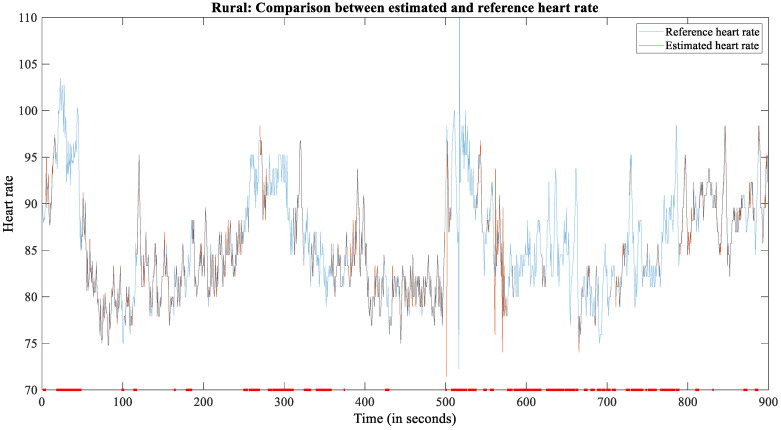
The plot of reference and estimated HR for subject 2209 during the rural scenario.

## Data Availability

The recorded dataset will be published with a CC BY-ND 4.0 license from the library of TU Braunschweig. The data are available under the following link: https://doi.org/10.24355/dbbs.084-202203170707-0, accessed on 28 May 2022.

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
