# Peer review of "Printed and Flexible ECG Electrodes Attached to the Steering Wheel for Continuous Health Monitoring during Driving"

_sensors, 2022, doi:10.3390/s22114198_

Round 1

Reviewer 1 Report

This paper presents a long-term monitoring of the electrocardiogram (ECG) during to earlier detection of symptoms of cardiovascular diseases. Many data collected from 19 subjects under four scenarios: rest, city, highway, and rural, total 950 minutes are provided to analyze the effectiveness of the proposed method. Overall, the topic of this paper is interesting. I have the following concerns:

  1. Abstract: It is better to highlight the major contribution of this work.
  2. Introduction: I think the authors missed a short summary of main contribution on your study, please highlight this. In addition, the outline of this paper should also be added.
  3. Introduction: Lots of references should be added into this part to enrich the context, such as many ECG signal processing methods can be used for detecting the symptoms of cardiovascular diseases [1-2]. In addition, many signal processing methods can also be used for analyzing the ECG signal [3-4]. Please add these references.

[1] "An Energy Efficient ECG Signal Processor Detecting Cardiovascular Diseases on Smartphone," in IEEE Transactions on Biomedical Circuits and Systems, vol. 11, no. 2, pp. 314-323, April 2017.

[2] "ECG signal processing for recognition of cardiovascular diseases: A survey," 2016 Sixth International Conference on Innovative Computing Technology (INTECH), 2016, pp. 677-682.

[3] "Target Detection Within Nonhomogeneous Clutter Via Total Bregman Divergence-Based Matrix Information Geometry Detectors," in IEEE Transactions on Signal Processing, vol. 69, pp. 4326-4340, 2021, doi: 10.1109/TSP.2021.3095725.

[4] "Adaptive Subspace Tests for Multichannel Signal Detection in Auto-Regressive Disturbance," in IEEE Transactions on Signal Processing, vol. 66, no. 21, pp. 5577-5587, 1 Nov.1, 2018.

4. It is better to provide some comparison results to verify your proposed methods. Please add more analysis about the results.

Author Response

  1. Reviewer

Comment:

This paper presents long-term monitoring of the electrocardiogram (ECG) during to earlier detection of symptoms of cardiovascular diseases. Many data collected from 19 subjects under four scenarios: rest, city, highway, and rural, total 950 minutes are provided to analyze the effectiveness of the proposed method. Overall, the topic of this paper is interesting. I have the following concerns:

  1. Abstract: It is better to highlight the major contribution of this work.

Response:

First, we want to thank you for the valuable feedback. We changed the abstract to highlight the major contribution (lines: 14-28):

“Continuous health monitoring in a vehicle enables earlier detection of symptoms of cardiovascular diseases. In this work, we design flexible and thin electrodes made of polyurethane for long-term monitoring of electrocardiogram (ECG) while driving. We determine the time of reliable ECG recording to evaluate the effectiveness of the electrodes. We record data from 19 subjects under four scenarios: rest, city, highway, and rural. The recording time is five min for rest and 15 min for the other scenarios. The total recording (950 min) is publicly available under a CC BY-ND 4.0 license. We use the simultaneous truth and performance level estimation (STAPLE) algorithm to detect the position of R-waves. Then, we derive the RR-interval to compare the estimated heart rate with the ground truth, which we obtain from ECG electrodes on the chest. We calculate the signal-to-noise ratio (SNR) and average it for the different scenarios. Highway has the lowest (SNR = -6.69 dB) and rural the highest (SNR = -6.80 dB), respectively. The usable time of the steering wheel is 42.46 % (city), 46.67 % (highway), and 47.72 % (rural). This indicates that steering wheel-based ECG recording is feasible and delivers reliable recordings from about 45.62 % of the driving time. In summary, the developed electrodes allow a continuous in-vehicle heart rate monitoring, and our publicly available recordings provide the opportunity to apply more sophisticated data analytics.”

Comment:

  1. Introduction: I think the authors missed a summary of the main contribution to your study, please highlight this. In addition, the outline of this paper should also be added.

Response:

We adjusted the chapter “Introduction” and added a summary of the main contribution as well as an outline of this paper (lines: 79-88):

“To investigate whether continuous health monitoring with steering wheel-based ECG recordings is feasible, we focus on designing electrodes and answering the question, "What percentage of the driving time results in reliable ECG signals?".

In Chapter 2, we explain the development process of the ECG electrodes, the sensor system for the recording, the experimental design, how a person can access the recorded data, and the generation of the ground truth. Chapter 3 present a comparison of the SNRs for different driving scenarios and between the steering wheel ground truth data. We discuss the applied methods and the results in Chapter 4. Our paper ends with a conclusion (Chapter 5).”

Comment:

  1. Introduction: Lots of references should be added into this part to enrich the context, such as many ECG signal processing methods can be used for detecting the symptoms of cardiovascular diseases [1-2]. In addition, many signal processing methods can also be used for analyzing the ECG signal [3-4]. Please add these references.

[1] "An Energy Efficient ECG Signal Processor Detecting Cardiovascular Diseases on Smartphone," in IEEE Transactions on Biomedical Circuits and Systems, vol. 11, no. 2, pp. 314-323, April 2017.

[2] "ECG signal processing for recognition of cardiovascular diseases: A survey," 2016 Sixth International Conference on Innovative Computing Technology (INTECH), 2016, pp. 677-682.

[3] "Target Detection Within Nonhomogeneous Clutter Via Total Bregman Divergence-Based Matrix Information Geometry Detectors," in IEEE Transactions on Signal Processing, vol. 69, pp. 4326-4340, 2021, doi: 10.1109/TSP.2021.3095725.

[4] "Adaptive Subspace Tests for Multichannel Signal Detection in Auto-Regressive Disturbance," in IEEE Transactions on Signal Processing, vol. 66, no. 21, pp. 5577-5587, 1 Nov.1, 2018.

Response:

Thank you for the suggestion of additional references to enrich the introduction with ECG signal processing methods. We added these references (lines: 65-77):

“Moreover, ECG signal processing methods already exist to recognize cardio-vascular diseases (CVD). In 2016, Ahmed et al. published a survey for ECG signal preprocessing [19]. They discuss CVD recognition based on preprocessing, QRS detection, feature extraction, and classification and pointed out that baseline wander and noise removal can be performed adaptive [20] or using median [21] and bandpass filters [22]. QRS detection often is based on derivatives [23], wavelet transform [24], neural networks [25], or hidden Markov models [26]. Ahmed et al. categorized the features for classification in time, frequency, and time frequency-based features [19]. In literature, the variety of classifiers reaches from neural networks [27] to a k-nearest neighbor [28]. In 2021, Hua et al. developed an approach for information divergences based on divergence-based matrix information geometry detectors [29]. Besides, Jain et al. implement an application-specific integrated circuit (ASIC) on a smartphone to detect symptoms energy-efficiently [30]. Furthermore, several work published multi-sensor or multi-channel approaches [31],[32].”

Comment:

  1. It is better to provide some comparison results to verify your proposed methods. Please add more analysis about the results.

Response:

The focus of this paper is on the development of thin and flexible electrodes, in-vehicle data recording, and classification of reliable and unreliable signal periods. To foster further analysis with machine learning or other rule-based algorithms, we published the recorded data. We stress this limitation in the discussion of our paper (lines: 223-227):

“Another limitation of our work is that we focus on the development of thin and flexible electrodes, in-vehicle data recording, and classification of reliable and unreliable signal periods but not on advanced signal analysis. To foster further data analytics with machine learning or other rule-based algorithms, our data is available https://publikationsserver.tu-braunschweig.de/receive/dbbs_mods_00070485.”

Reviewer 2 Report

This paper did an interesting work on monitoring ECG during driving by using flexible ECG electrodes on steering wheel. All the tests and results are clearly given with a well-written English. I think the paper can be accepted after dealing with some minor issues.

  1. As the authors claimed, the humidity may influence the measuring performances. However, sweat of hands may appears after a longer driving time. This induces two problems: i) a comparison between the SNRs at the beginning and a long driving time (with a humidity information is better); ii) air breathability of electrodes. More sweat may accumulate between hand and electrodes after a longer driving, which may induce a larger error in the results. Some discussions about the two problems can be helpful.

2.The rules for determining the signals usable or unusable are needed in the paper.

3. The signal reliability of 2209 is given. Is it the lowest value in the tests? What about the results for the whole test?  

Author Response

  1. Reviewer

Comment:

This paper did an interesting work on monitoring ECG during driving by using flexible ECG electrodes on the steering wheel. All the tests and results are given with a well-written English. I think the paper can be accepted after dealing with some minor issues.

  1. As the authors claimed, the humidity may influence the measuring performances. However, sweat of hands may appears after a longer driving time. This induces two problems: i) a comparison between the SNRs at the beginning and a long driving time (with a humidity information is better); ii) air breathability of electrodes. More sweat may accumulate between hand and electrodes after a longer driving, which may induce a larger error in the results. Some discussions about the two problems can be helpful.

Response:

Thank you very much for spending your time and giving us your valuable feedback. We added this point in chapter “Discussion” (lines: 238-242):

“In our calculations for the SNR, we could not identify a change of the SNR between the beginning and end of a longer driving time. This could occur if the sweat accumulates after a long driving from the hands on the electrodes. However, we expect a better signal on sweating hands, as the sweat is composed of electrolytes and increases the conductivity [54].”

Comment:

  1. The rules for determining whether the signals are usable or unusable are needed in the paper.

Response:

Thanks for your suggestion. We complemented the new chapter “Rules for Classification in Usable and Unusable Signal” (lines: 159-166):

“For the preprocessing, we use a Butterworth bandpass (Fig 3) to remove baseline wander and noise with the cut-off frequencies of 0.5 Hz and 25 Hz [46]. We consider the RRI as a time-domain feature. We classify signal segments as usable based on two thresholds. The first considers the median of the reference HR, which forms a tube around the signal to exclude too high amplitudes. The normal range for HR is between 60 bpm and 100 bpm [47]. Based on Langendorf et al. [35], we label a segment as unreliable if the RRI is changing more than 0.08 s.”

Comment:

  1. The signal reliability of 2209 is given. Is it the lowest value in the tests? What about the results for the whole test?

Response:

In chapter “Results”, we outlined our selection of subject 2209 (lines: 184-186):

“In the following figures, we analyze the recordings of test person 2209, arbitrarily choosen as an average example. The SNRs from this test person are 0.84 dB (rest), -7.25 dB (city), -7.78 dB (highway) and -6.99 dB (rural) and neither the worst nor the best values.”

Reviewer 3 Report

The manuscript is well written and provides insight into how to improve the ability to monitor heart performance while driving.  The authors believe that reliable data can be extracted during driving.  One question would be;

  • Have the authors considered using machine learning to track the data over extended periods of time to see if heart performance has changed from a baseline? This might help flag anomalies more conclusively. 

Author Response

  1. Reviewer

Comment:

The manuscript is well written and provides insight into how to improve the ability to monitor heart performance while driving.  The authors believe that reliable data can be extracted during driving.  One question would be;

Have the authors considered using machine learning to track the data over extended periods of time to see if heart performance has changed from a baseline? This might help flag anomalies more conclusively.

Response:

We thank you for your valuable feedback as it improved our work.

It is a valid point to use a machine learning algorithm for data analysis. The scope of this paper is to develop new electrodes and to determine the data quality.  Therefore, we added the point of usage of a machine learning algorithm in the chapter “discussion” (lines: 227-231):

“The usage of machine learning approaches, e.g., support vector machines, k-nearest neighbor, and convolutional neural networks (CNN), could lead to improved data analysis. This integration could increase the usable recording time and allows data tracking over extended periods of time to see if heart performance has changed from a baseline.”

Round 2

Reviewer 1 Report

The authors have addressed all my concerns.

Reviewer 2 Report

The paper is well revised, and it can be accepted now.